# DIVING INTO OPTIMIZATION OF TOPOLOGY IN NEURAL NETWORKS

## ABSTRACT

Seeking effective networks has become one of the most crucial and practical areas in deep learning. The architecture of a neural network can be represented as a directed acyclic graph, whose nodes denote the transformation of layers and edges represent information flow. Despite the selection of *micro* node operations, *macro* connections among the whole network, noted as *topology*, largely affect the optimization process. We first rethink the residual connections via a new *topological view* and observe the benefits provided by dense connections to the optimization. Motivated by this, we propose an novel method to optimize the topology of a neural network. The optimization space is defined as a complete graph, and through assigning learnable weights which reflect the importance of connections, the optimization of topology is transformed into learning a set of continuous variables of edges. To extend the optimization to larger search spaces, a new series of networks, called TopoNet, are designed. To further focus on critical edges and promote generalization ability in dense topologies, an auxiliary sparsity constraint is adopted to constrain the distribution of edges. Experiments on classical networks prove the effectiveness of the optimization of topology. Experiments with TopoNets further verify both availability and transferability of the proposed method in different tasks e.g. image classification, object detection, and face recognition.

## 1 INTRODUCTION

Deep learning successfully transits the feature engineering from manual to automatic design. As a tendency, seeking effective neural networks gradually becomes an important and practical direction. Topologically, the architecture of a network can be expressed as a direct acyclic graph, whose nodes denote the transformation of layers and edges represent information flow. Corresponding to *micro* operations within a layer/block, the *macro* connections (Atwood & Towsley, 2016; Pérez-Rúa et al., 2018) between layers also experienced a series of evolutions. And we denote the *macro* connections as *topology* in this work.

In initial literature, AlexNet (Krizhevsky et al., 2012), VGGNet (Simonyan & Zisserman, 2014) with *plain* topology were proposed. Due to the problems of gradient vanishing and exploding, extending the network to a deeper level for better representation is nearly difficult. To better adapt the optimization process of gradient descent, GoogleNet (Szegedy et al., 2015) adopted parallel modules, and Highway networks (Srivastava et al., 2015) utilized gating units to regular the flow of information, resulting in *multipath* and *elastic* topologies. Driven by the significance of depth, the residual block consisted of residual mapping and shortcut was raised in ResNet (He et al., 2016). Topological changes in neural networks successfully scaled up neural networks to hundreds or even thousands of layers. The *residual* topology was widely approved and applied in the following works, e.g. MobileNet (Sandler et al., 2018; Howard et al., 2019) and ShuffleNet (Zhang et al., 2018). Divergent from relative sparse topologies, DenseNet (Huang et al., 2017) wired densely within a block to reuse features fully. Recent advances in computer vision also explore neural architecture search (NAS) methods (Zoph et al., 2018; Liu et al., 2019b; Tan et al., 2019) to jointly search *micro* operations and topology. To trade-off efficiency and performance, their topologies are hand-designed stacked patterns and constrained in limited searching spaces. Orthogonally, randomly wired networks (Xie et al., 2019) explored *random* topology as architecture by different generators, and obtained competitive performance. These trends reflect the great impact of topology on the

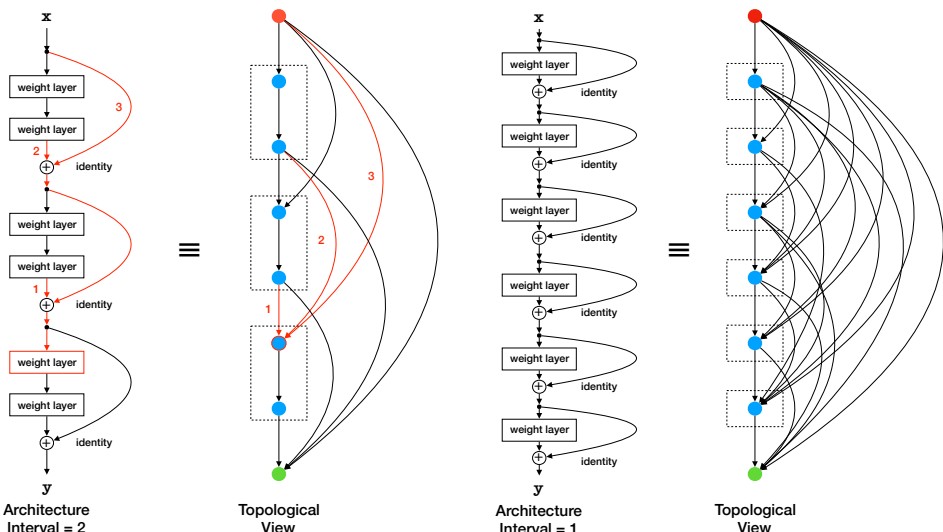

Figure 1: From natural view to topological view of networks with residual connections. We map both addition and unary layer to nodes, and data flow to edges. Red node denotes input **x** and green one means output **y**. Red arrows give an example of this mapping for a node with in-degree of 3.

optimization of neural networks. Echoing this perspective, we wonder: *Can topology itself in neural networks be optimized? What is the suitable route to do this?*

To answer these questions, we start by understanding the most representative residual connection in terms of topology. It formulates $x + \mathcal{F}(x)$ as a basic component, in which $x$ represents the identity shortcut, and $\mathcal{F}(x)$ denotes the residual mapping. Figure 1 presents two residual architectures with a interval of 1 and 2 respectively. And the number of intervals represents the number of layers that make up a block. If we map both combining (e.g., addition) and transformation (e.g., $3 \times 3$ conv) to a node, and flow of features to an edge, then the architecture can be expressed as a directed acyclic graph (DAG). From the *topological view*, the residual connections are rather denser than the natural perspective. This novel representation illustrates residual networks perform multiple feed-forward paths instead of a single deep network. When the interval degrades to 1, its topology evolves into a complete DAG. For a complete DAG with $n$ nodes, the number of available paths from input to output is $(n-1)!$. As a result, most layers are directly connected to the input and output, resulting in direct access to the gradients and the original input. So we make a hypothesis that *dense wirings in topology benefit the optimization process*.

To verify the hypothesis and explore the effects of differently wired topologies, toy experiments on CIFAR-100 (Krizhevsky et al., 2009) are conducted. To ensure universality, three types of layer are selected respectively, including $3 \times 3$ conv, $3 \times 3$ inverted bottleneck (Sandler et al., 2018) and $3 \times 3$ separable depthwise conv (Howard et al., 2017). Defining the total number of nodes, we establish three series of networks under different intervals. Loss curves during training and test performance are given in Figure 2. It demonstrates an obvious topology-induced impact on optimization. First, it shows denser networks (intervals of 1 and 2) can achieve lower training losses in the majority of cases. Second, the optimal topology is different for different node types (2 for conv, 1 for depthwise and inverted bottleneck). It is possible due to that not all connections are beneficial, and the aggregation of features should be discretionary. So we modify the hypothesis as the *topology with dense and effective connections are necessary for networks that are compatible with the optimization process*. Regardless of the node types, the optimal structure is always a subset of the complete graph. Naturally, finding the optimal topology in a full search space is equivalent to finding the optimal sub-graph in the complete graph.

To focus on the optimization of topology itself, in this work, we exclude the influence of the mixture of different layers/nodes and select the complete DAG as the search space. Through assigning learnable weights which reflect the importance of connections of edges, the topology can be optimized

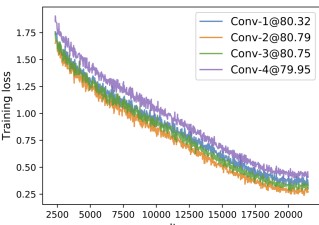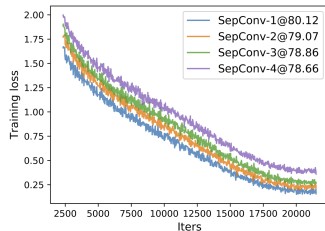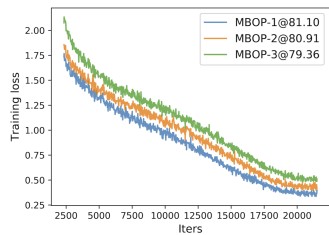

Figure 2: The impact of topology changes on the training process in CIFAR-100. Each subgraph represents the loss curves of networks composed of a single node type when the topology changes. The tags at the top give the *type* of node used, *interval* of residual blocks and test Top-1 *accuracy*.

through gradient descent. Then task-related architectures can be achieved conveniently. Besides, this optimization method is also compatible with existing networks. Given a basic network, optimization of topology can lead to improvement without additional computing burdens. To verify the effectiveness of the optimization of topology in larger search spaces, we also propose a well-designed network named TopoNet. To solve the optimization problem in a large space and promote generalization, we attach auxiliary sparsity regularization to the distribution of edges, resulting in critical connections and better generalization. Our contributions are as follows:

- We first rethink the residual connections via a new *topological view* and observe the benefits provided by dense connections to the optimization.
- We propose an innovative method to optimize the topology of a neural network. The optimization space is defined as a complete graph, through assigning learnable weights which reflect the importance of connections, the optimization of topology is transformed into learning a set of continuous variables of edges.
- The proposed optimization method of topology is effective for existing networks.
- When extended to larger search space, TopoNets and optimization using sparsity constraint verify the availability and transferability in different tasks, including image classification on ImageNet, object detection on COCO and face recognition on MegaFace.

## 2 METHODOLOGY

### 2.1 TOPOLOGICAL REPRESENTATION OF ARCHITECTURE

We represent the neural network using a directed acyclic graph (DAG) in topology: $\mathcal{G} = (\mathcal{N}, \mathcal{E})$. Each node $n_i \in \mathcal{N}$ owns an operation $o_i$, such as convolution or identity mapping, parametrized by $\mathbf{w}_i$, where $i$ stands for the topological ordering. Edge $e_{ji} = (j, i, \alpha_{ji}) \in \mathcal{E}$ means the flow of features from node $j$ to node $i$, and the importance of the connection is determined by $\alpha_{ji}$. Each node aggregates inputs from connected pre-order nodes, and performs a transformation to get an output tensor $\mathbf{x}_i$. Then $\mathbf{x}_i$ is sent out to the post-order nodes by the output edges. Among these nodes, the first in topological ordering is appointed to be input, and the last to be output. These two nodes do not transform features, just send/gather features. Except for the input and output nodes, other nodes perform feature transformation, such as convolution, activation and normalization. To focus on the impact of topological changes, the same type of operation is applied for all nodes. It can be formulated as follows:

$$\mathbf{x}_i = o_i(\mathbf{x}'_i; \mathbf{w}_i), \text{ where } \mathbf{x}'_i = \sum_{e_{ji} \in \mathcal{E}} \alpha_{ji} \cdot \mathbf{x}_j. \tag{1}$$

For a network with $k$ stages, $k$ DAGs are initialized and connected in series. Each graph is linked to its preceding or succeeding stage by output or input node. The topology of network can be denoted as a set $\boldsymbol{\alpha} = \{\alpha_{ji}^k\}$, and weights of transformation as a set $\mathbf{w} = \{\mathbf{w}_i^k\}$. Given an input $\mathbf{x}$, the mapping function from sample to feature can be written as:

$$\mathcal{F}(\mathbf{x}) = \mathcal{T}^k(\cdots \mathcal{T}^2(\mathcal{T}^1(\mathbf{x}; \boldsymbol{\alpha}^1, \mathbf{w}^1); \boldsymbol{\alpha}^2, \mathbf{w}^2) \cdots; \boldsymbol{\alpha}^k, \mathbf{w}^k), \tag{2}$$

where $\mathcal{T}(\cdot)$ denotes the mapping function computing feature maps of $\mathcal{G}$.

Table 1: Architectures of TopoNets for ImageNet.

| Layers | Output Size | Component and Channels | Edges of Different Topologies | | |
|--------|-------------|------------------------|-------------------------------|---|---|
| | | | Random, $p$ | Residual, $l$ | Complete |
| Head | $112 \times 112$ | $7 \times 7$ conv, $C$ | - | - | - |
| Stage 1 | $56 \times 56$ | $N_1$ nodes, $2 \times C$ | $p \cdot \binom{N_1}{2}$ | $\frac{N_1-2}{l} + \binom{\frac{N_1-2}{l}+2}{2}$ | $\binom{N_1}{2}$ |
| Stage 2 | $28 \times 28$ | $N_2$ nodes, $4 \times C$ | $p \cdot \binom{N_2}{2}$ | $\frac{N_2-2}{l} + \binom{\frac{N_2-2}{l}+2}{2}$ | $\binom{N_2}{2}$ |
| Stage 3 | $14 \times 14$ | $N_3$ nodes, $8 \times C$ | $p \cdot \binom{N_3}{2}$ | $\frac{N_3-2}{l} + \binom{\frac{N_3-2}{l}+2}{2}$ | $\binom{N_3}{2}$ |
| Stage 4 | $7 \times 7$ | $N_4$ nodes, $16 \times C$ | $p \cdot \binom{N_4}{2}$ | $\frac{N_4-2}{l} + \binom{\frac{N_4-2}{l}+2}{2}$ | $\binom{N_4}{2}$ |
| Classifier | $1 \times 1$ | GAP, 1k-d FC, $\mathrm{softmax}$ | - | - | - |

## 2.2 SEARCH SPACE AND OPTIMIZATION METHOD

In most previous literature (Tan et al., 2019; Guo et al., 2019), searching for a cell that is later stacked as blocks for a deep network is an expedient solution to trade-off search efficiency and result optimality. But it constrains the possible topologies in a small subset of all possible graphs. Different from them, we relax connections from block-wise to layer-wise, and define the search space as a complete DAG. It provides all possible connections and is much wider than cell-based or block-based approaches. For a complete DAG with $N$ nodes, the search space contains $2^{N(N-1)/2}$ possible topological structures. Assigning learnable weights to edges transforms the optimization of topology into learning a set of continuous variables weights for edges $\boldsymbol{\alpha}$. It provides a differentiable type to update topology through gradients according to the loss of task noted as $\mathcal{L}_{task}$. Compared with existing differentiable methods (e.g. DARTS), there is no need to conduct $\arg\max$ operation for $\boldsymbol{\alpha}$ since it determines the fusion of features instead of the selection of operations. Continuous $\boldsymbol{\alpha}$ also guarantees the consistency between training and testing. During the process of optimization, both $\mathbf{w}$ and $\boldsymbol{\alpha}$ are optimized jointly. This optimization process can be viewed as:

$$\min_{\mathbf{w},\boldsymbol{\alpha}} \mathcal{L}_{task} = \min_{\mathbf{w},\boldsymbol{\alpha}} \mathcal{L}(\mathcal{F}(\mathbf{x}; \mathbf{w}, \boldsymbol{\alpha}), \mathbf{y}) \qquad (3)$$

## 2.3 EXPANDING TO LARGER SEARCH SPACES BY TOPONET

The verification of the optimization method requires a network as a base, which determines the upper bound of the search space. Existing networks are limited by the number of nodes, and can only provide narrow possible structures. These may cover the influence caused by topological changes and affect the search for optimal topology.

Therefore, we design a series of architectures named as TopoNets that can flexibly adjust search space, types of topology and node. As shown in Table 1, it consists of four stages with number of nodes of $\{N_1, N_2, N_3, N_4\}$. The topology in each stage is defined by a graph, whose type can be chosen from {*complete, random, residual*}. The *complete* graph is used for the optimization of topology. For a more strict comparison, we also take the other two types as baselines. The *residual* one has been described in detail before, and the interval is referred to $l$. In the *random* one, an edge between two nodes is linked with probability $p$, independent of all other nodes and edges. The higher the probability, the denser it is. We follow two simple design rules used in (He et al., 2016), (i) in each stage, the nodes have the same number of filters $C$; (ii) and if the feature map size is halved, the number of filters is doubled. The change of filters is implied by the first calculation node in each graph. For the head of the network, we use a single convolutional layer for simplicity. The network ends with a classifier composed of a global average pooling (GAP), a 1000-dimensional fully-connected (FC) layer and $\mathrm{softmax}$ function.

## 2.4 SPARSITY CONSTRAINT

Much as the mammalian brain (Rauschecker, 1984) in biology, where synapses are created in the first few months of a child's development, followed by gradual re-weighting through postnatal knowledge, growing into a typical adult with relative sparse connections. Same observation also be found in Figure 2 (left) that moderate sparsity benefits the generalization. Inspired from these, we choose

---

**Algorithm 1:** Optimization of Topology in Neural Networks.

---

**Input:** Network wired with $k$ DAGs $\mathcal{G}(\mathcal{N}, \mathcal{E}; \mathbf{w}, \boldsymbol{\alpha})$; training set $\{(\mathbf{x}, \mathbf{y})^{(s)}\}_{s=1}^{S}$; *SparseType*; $\lambda$.
**Output:** Optimized weights $\mathbf{w}$ and topology $\boldsymbol{\alpha}$.
**for** $s = 1 \ldots S$ **do**
    Conduct forward propagation $\mathcal{F}(\mathbf{x})$, obtain $\mathcal{L}_{task}$ and $\mathcal{L}_{sparse}$ ;
    Calculate backward gradient and update $\mathbf{w}$ with $\nabla_{\mathbf{w}} \mathcal{L}_{task}$ ;
    **if** *SparseType* **is** *uniform* **then**
        Update $\boldsymbol{\alpha}$ with $\nabla_{\boldsymbol{\alpha}} \mathcal{L}_{task} + \lambda \cdot \nabla_{\boldsymbol{\alpha}} \mathcal{L}_{sparse}$ ;
    **else if** *SparseType* **is** *adaptive* **then**
        Count input edges $\{e_{ji}|j < i\}$ and its in-degree $\delta_i$ for each node $n_i$;
        Update $\alpha_{ji}$ according to $\delta_i$ with $\nabla_{\alpha_{ji}} \mathcal{L}_{task} + \lambda \cdot \log(\delta_i) \cdot \nabla_{\alpha_{ji}} \mathcal{L}_{sparse}$ ;
**end**

---

L1 regularization, denoted as $\mathcal{L}_{sparse}$, to penalize non-zero parameters of edges resulting in more parameters near zero. This sparsity constraint promotes attention to more critical connections. Then the objective function of our proposed method can be formulated as:

$$\min_{\mathbf{w}, \boldsymbol{\alpha}} \mathcal{L}_{task} + \lambda \cdot \mathcal{L}_{sparse} = \min_{\mathbf{w}, \boldsymbol{\alpha}} \mathcal{L}(\mathcal{F}(\mathbf{x}; \mathbf{w}, \boldsymbol{\alpha}), \mathbf{y}) + \lambda \cdot \|\boldsymbol{\alpha}\|_1, \tag{4}$$

where $\mathbf{y}$ stands for the label w.r.t the input $\mathbf{x}$, and $\lambda$ is a hyper-parameter to balance the sparsity level. During the process of back-propagation, $\mathbf{w}$ can be updated through $\nabla_{\mathbf{w}} \mathcal{L}_{task}$. Due to the properties of a complete graph, we propose two types to update $\boldsymbol{\alpha}$. The first one attaches sparsity constraint on all weights of edges uniformly by computing $\nabla_{\boldsymbol{\alpha}} \mathcal{L}_{task} + \lambda \cdot \nabla_{\boldsymbol{\alpha}} \mathcal{L}_{sparse}$, and is called *uniform sparse*. The other one is logarithmically related to the in-degree $\delta_i$ of a node $n_i$ with input edges $\{e_{ji}|j < i\}$. It performs larger constraint on dense input and smaller on sparse input, and is noted as *adaptive sparse*. These two types will be further discussed in the experiments section. Algorithm 1 summarizes the procedure for the optimization of topology in neural networks.

## 3 EXPERIMENTS AND ANALYSIS

### 3.1 OPTIMIZATION FOR CLASSICAL NETWORKS

To investigate the applicability of the optimization method on representative networks, we select ResNet-CIFAR (He et al., 2016) consisted of $3 \times 3$ *conv* and MobileNetV2-1.0 (Sandler et al., 2018) consisted of *inverted bottleneck* as node operation. For the optimization of ResNets, we change the interval of 2 in the BasicBlock to 1 to form the complete graph. For MobileNetV2-1.0, each node involves a residual connection and can be viewed as complete graph naturally. As a complement, we increase the depth by increasing the node in each stage, resulting in larger search spaces. It is also a common skill to expand networks (Tan & Le, 2019). Through assigning learnable $\boldsymbol{\alpha}$ to their edges, the topologies can be optimized using our method.

Table 2: Top-1 accuracy (%) of networks on CIFAR100.

| Basic Network | Params(M) | FLOPs(G) | Original | Optimized | Gain |
|---|---|---|---|---|---|
| ResNet-20 | 0.28 | 0.04 | 69.01 | $69.91 \pm 0.12$ | 0.90 |
| ResNet-32 | 0.47 | 0.07 | 72.07 | $73.34 \pm 0.09$ | 1.37 |
| ResNet-44 | 0.67 | 0.10 | 73.73 | $75.60 \pm 0.14$ | 1.87 |
| ResNet-56 | 0.86 | 0.13 | 75.22 | $76.90 \pm 0.03$ | 1.68 |
| ResNet-110 | 1.74 | 0.25 | 76.31 | $78.54 \pm 0.15$ | 2.23 |

We evaluate our method on CIFAR100 and ImageNet 2012 classification dataset (Russakovsky et al., 2015) that consists of 100 and 1000 classes respectively. Comparative experiments are shown in Table 3. Under similar Params and FLOPs, the optimization of topology brings 2.22% improvement on Top-1 for ResNet-110 on CIFAR100. For MobileNetV2-1.0 and its extension, the improvements range from 0.24% to 0.75% separately. These confirm the optimization of topology is effective to existing networks and depth-friendly, and larger search space can bring more improvements.

A detailed experimental setup for this section can be found in Sect. A.1.

Table 3: Top-1 accuracy (%) of scaled networks on ImageNet.

| Basic Network | Params(M) | FLOPs(G) | Original | Optimized | Gain |
|---|---|---|---|---|---|
| MobileNetV2-1.0 | 3.51 | 0.31 | 72.60 | $72.86 \pm 0.13$ | 0.24 |
| MobileNetV2-1.0-2N | 6.43 | 0.62 | 75.93 | $76.40 \pm 0.05$ | 0.47 |
| MobileNetV2-1.0-4N | 9.62 | 1.10 | 77.33 | $77.87 \pm 0.09$ | 0.54 |
| MobileNetV2-1.0-6N | 12.00 | 2.06 | 77.61 | $78.36 \pm 0.14$ | 0.75 |

## 3.2 OPTIMIZATION FOR DESIGNED TOPONETS

Due to restricted optional topologies of classical networks, the topology can be only optimized in small search spaces, which limits the representation ability of topology. In this section, we design a larger search space using TopoNets, and fully illustrate the improvement brought by topology optimization. The properties of edges and nodes in the optimized topology are also analyzed.

### 3.2.1 EXPERIMENTAL SETUP FOR TOPONET

To initialize as many nodes as possible with defined parameters and calculations, we select the separable depthwise convolution that includes a $3 \times 3$ depthwise convolution followed by a $1 \times 1$ pointwise convolution. Similar with (Xie et al., 2019), a triplet unit ReLU-conv-BN forms the transformation in a node. The number of nodes in each stage is $\{14, 20, 26, 14\}$. In this setting, the number of possible discrete topologies is $2^{\binom{14}{2}} \cdot 2^{\binom{20}{2}} \cdot 2^{\binom{26}{2}} \cdot 2^{\binom{14}{2}} \approx 6 \times 10^{209}$. The weights of $\alpha$ are initialized to be 1. And $C$ is set to be 64 to match ResNet-50 (He et al., 2016) for equivalent comparison in the amount of parameters and calculations.

Table 4: Optimization results for designed TopoNets on ImageNet.

| Network | Initial Edges | Params(M) | FLOPs(G) | Top-1(%) |
|---|---|---|---|---|
| ResNet-50 | - | 25.57 | 4.08 | 76.50 |
| *Random*, $p = 0.8$ | {56,130,221,56} | 23.23 | 3.95 | $77.56 \pm 0.22$ |
| *Random*, $p = 0.6$ | {44,104,172,44} | 23.23 | 3.95 | $77.84 \pm 0.19$ |
| *Random*, $p = 0.4$ | {31,64,117,31} | 23.23 | 3.95 | $77.90 \pm 0.27$ |
| *Residual*, $l = 4$ | {19,36,46,19} | 23.23 | 3.95 | $77.72 \pm 0.13$ |
| *Residual*, $l = 3$ | {23,40,61,23} | 23.23 | 3.95 | $78.10 \pm 0.07$ |
| *Residual*, $l = 2$ | {34,64,103,34} | 23.23 | 3.95 | $78.26 \pm 0.04$ |
| *Complete* | {91,190,325,91} | 23.23 | 3.95 | $77.24 \pm 0.12$ |
| *Complete*, $\alpha$ | {91,190,325,91} | 23.23 | 3.95 | $78.22 \pm 0.13$ |
| *Complete*, $\alpha$, *uniform sparse* | {91,190,325,91} | 23.23 | 3.95 | $78.46 \pm 0.04$ |
| *Complete*, $\alpha$, *adaptive sparse* | {91,190,325,91} | 23.23 | 3.95 | $78.60 \pm 0.06$ |

†The changes of Params and FLOPs caused by edges for different topologies are negligible.

### 3.2.2 IMAGE CLASSIFICATION

We conduct experiments in image classification as in Sect. 3.1. The validation results are shown in Table 4. Some conclusions can be drawn from the results. (i) Compared with ResNet-50, TopoNets achieve higher performances overall. These reflect the efficiency and elegance of the designed network. (ii) TopoNets of *random (0.4 ∼ 0.8)* and *residual (2 ∼ 4)* reflect the benefits of moderate sparsity using prior design. (iii) Under *complete* graphs, direct optimization of topology can yield $0.98\%$ improvement on Top-1. A similar result is obtained by *residual (2)*, which further reflects residual connection is a pretty manual-designed topology. (iv) Through assigning sparsity constraints, performances have been further improved. TopoNet of *adaptive sparse* gets the best Top-1 by $78.61\%$. This proves the benefits of sparseness for dense topology.

In order to intuitively understand the optimization effect of sparsity constraints on dense topological connections, distributions of $\alpha$ in optimized topologies are given in Figure 3. Compared with *w/o sparse*, sparsity constraints push more parameters near zero, resulting in focusing on critical con-

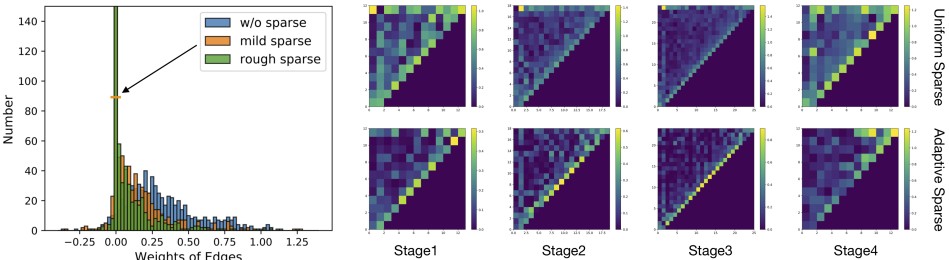

Figure 3: The effect of sparsity constraint on distributions of $\boldsymbol{\alpha}$. Histogram on the left indicates that sparsity drives most of the weights near zero. Adjacency matrices on the right shows the difference between *uniform* and *adaptive* one, whose rows correspond to the input edges for a particular node and columns represent the output ones. Colors indicate the weights of edges.

nections. With the enhancement of constraints, more connections disappear. Excessive constraints will damage the representation of features, so $\lambda$ is set to be e-4 in all experiments. In the right, two types of constraints are shown in different stages. The *adaptive* one penalizes denser connections a lot and further strengthens relatively sparser but critical connections.

A detailed experimental setup for this section can be found in Sect. A.2.1.

### 3.2.3 GRAPH DAMAGE FOR TOPOLOGICAL PROPERTIES

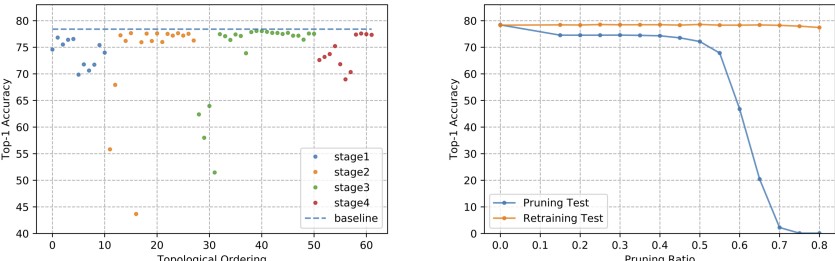

Figure 4: Impact of node (left) and edge (right) removal for *complete* type of TopoNet.

To analyze the optimized topology, we select the trained *complete* model in Table 4.

First, we explore the influence by removing individual nodes according to its topological ordering in the graph and evaluating without extra training. We expect them to break because dropping any layer drastically changes the input distribution of all subsequent layers. Surprisingly, most removals do not lead to a noticeable change as shown in Figure 4 (left). It can be explained that the available paths is reduced from $(n-1)!$ to $(n-2)!$, leaving sufficient paths. This suggests that each node in the complete graph do not strongly rely on others, although it is trained jointly. Direct links with input/output nodes makes each node contribute to the final feature representation, and benefits the optimization process. Another observation is that nodes in the front of topological orderings contribute more to a stage. This can be explained that for a node with ordering of $i$, the generated $x_i$ can be only received by node $j$ (where $j > i$). This causes the feature generated by the front nodes can participate in aggregation as a downstream input. It makes the front nodes contribute more. This can be used to reallocate calculation resources in future work.

Second, we consider the impact of edge removal. All edges with $\alpha$ below a threshold are pruned from the graph, only remaining the important connections. Accuracies before and after retraining are given in Figure 4 (right). Without retraining, accuracy decreases as the degree of pruning deepens. It is interesting to see that we have the "free lunch" of reducing less than 40% without losing much accuracy. If we fix $\alpha$ of remaining edges and retrain the weights, it can maintain accuracy with 80% of the nodes removed. This proves that the optimization process has found indeed important

connections. After pruning edges, nodes with zero in-degree or zero out-degree may be safely removed. It can be used to reduce the parameters and accelerate inference in practical applications.

### 3.2.4 TRANSFERABILITY ON DIFFERENT TASKS

To evaluate the generalization and transferability for both optimization method and TopoNets, we also conduct experiments on COCO object detection task (Lin et al., 2014). We adopt FPN (Lin et al., 2017) as the object detection method. The backbone is replaced with corresponding pretrained one in Table 4, and is fine-tuned on COCO train2017 dataset. The training configurations of different models are consistent. Test performances are given in Table 5. To comparable ResNet-50, TopoNets obtain significant promotions in AP with lower computation costs. Contrast with elegant residual topology, our optimization method can also achieve increase by $0.95\%$. These results indicate the effectiveness of the proposed network and the optimization method.

A detailed experimental setup for this object detection can be found in Sect. A.2.2. Our method also succeeds in face recognition, whose details are given in Sect. A.2.3.

Table 5: Transferability results (%) on COCO object detection.

| Backbone | AP | $AP_{50}$ | $AP_{75}$ | $AP_S$ | $AP_M$ | $AP_L$ |
|---|---|---|---|---|---|---|
| ResNet-50 | 36.42 | 58.66 | 38.90 | 21.93 | 39.84 | 46.74 |
| *Residual*, $l = 2$ | 40.74(+4.32) | 63.22 | 44.62 | 25.01 | 44.18 | 52.74 |
| *Complete*, $\alpha$ | 41.35(+4.93) | 63.32 | 45.08 | 25.63 | 44.99 | 53.47 |
| *Complete*, $\alpha$, *uniform sparse* | 41.46(+5.04) | 63.83 | 44.91 | 25.07 | 45.31 | 53.52 |
| *Complete*, $\alpha$, *adaptive sparse* | 41.69(+5.27) | 63.86 | 45.45 | 25.58 | 45.52 | 53.69 |

## 4 RELATED WORKS

We briefly review related works in the aspects of neural network structure design and relevant optimization methods.

Neural network design is widely studied in previous literature. From shallow to deep, the shortcut connection plays an important role. Before ResNet, an early practice (Venables & Ripley, 2013) also added linear layer connected from input to the output to train multi-layer perceptrons. Besides, "inception" layer was proposed in (Szegedy et al., 2015) that is composed of a shortcut branch and a few deeper branches. Except on large networks, shortcut also proved effective in small networks, e.g. MobileNet (Sandler et al., 2018), ShuffleNet (Zhang et al., 2018) and MnasNet (Tan et al., 2019). The existence of shortcut eases vanishing/exploding gradients (He et al., 2016; Srivastava et al., 2015). In this paper, we explain from a *topological view* that shortcuts offer dense connections and benefit optimization. On the macro structure, there also exist many networks with dense connections. DenseNet (Huang et al., 2017) contacted all preceding layers and passed on the feature maps to all subsequent layers in a block. HRNet (Sun et al., 2019) benefits from dense high-to-low connections for fine representations. Densely connected networks promote the specific task of localization (Tang et al., 2018). Moreover, we optimize the desired network from the complete graph. This is different from MaskConnect (Ahmed & Torresani, 2018) which constrained by $K$ discrete in-degree. This also provides a supplement to (Xie et al., 2019) where random graphs generated by different generators are employed to form a network.

Above the optimization process, our method is consistent with DARTS (Liu et al., 2019b) which is differentiable. Different from it, we do not adopt alternative optimization strategies for weights and architecture. Joint training can replace the transferring step from one task to another, and obtain task-related topology. Different from sample-based optimization methods (Real et al., 2019), our method obtains optimized architecture in the manner of one-shot. (Bender et al., 2018; Guo et al., 2019) also explored this type and utilized weight-sharing across models to amortize the cost of training. Searching from the full space is evaluated in object detection by NAS-FPN (Ghiasi et al., 2019), in which the feature pyramid is sought in all cross-scale connections. In the aspect of semantic segmentation, Auto-DeepLab (Liu et al., 2019a) also formed a hierarchical architecture to enlarge possible spaces. The sparsity optimization can also be observed in other applications, e.g. path

selection for a multi-branch network (Huang & Wang, 2018), and pruning unimportant channels for fast inference (Han et al., 2015).

## 5 CONCLUSION

In this work, we have found a feasible way for the optimization of topology in neural networks. Motivated by the *topological view*, the optimization space is defined as a complete graph, through assigning learnable weights which reflect the importance of connections, the optimization of topology is transformed into learning a set of continuous variables of edges. This method is compatible with existing networks, such as ResNet and MobileNet. TopoNet and corresponding sparsity constraint also proved the effectiveness and transferability in the larger optimization space. Moreover, the observed properties of topology can be used for future works.

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

## A  APPENDIX

### A.1  EXPERIMENTAL DETAILS FOR CLASSICAL NETWORKS

#### A.1.1  RESNET-CIFAR

On CIFAR100, we train ResNets using 2 GPUs. [1] We follow the hyperparameter settings in paper (DeVries & Taylor, 2017), which is init lr = 0.1 divide by 5 times at 60th, 120th, 160th epochs. We train for 200 epochs with batchsize 128 and weight decay 5e-4. Nesterov (Sutskever et al., 2013) momentum of $0.9$ without dampening is also used. The training and test size is $32 \times 32$. We report classification accuracy on the validation set.

#### A.1.2  MOBILENETV2

On ImageNet, The models are trained on the 1.28 million training images and evaluated on the 50k validation images. We train MobileNetV2 using 16 GPUs for 200 epochs with batch size of 1024. The initial learning rate is 0.4 and cosine shaped learning rate decay (Loshchilov & Hutter, 2017) is adopted. Following (Sandler et al., 2018), we use a weight decay of 4e-5 and dropout (Hinton et al., 2012) of 0.2. Nesterov momentum of 0.9 without dampening is also used. The training and test size is $224 \times 224$. We report classification accuracy on the validation set. And the data augmentation is used the same as Sect. A.1.1.

### A.2  EXPERIMENTAL DETAILS FOR TOPONETS

#### A.2.1  IMAGE CLASSIFICATION

On ImageNet, TopoNets of *random*, *residual* and *complete* are trained using 16 GPUs for 100 epochs with batch size of 1024. The initial learning rate is $0.4$, and cosine shaped learning rate decay is adopted. We use a weight decay of e-4 and a Nesterov momentum of $0.9$ without dampening. Dropout is not used. Label smoothing regularization (Szegedy et al., 2017) with a coefficient of 0.1 is also used. The training and test size is $224 \times 224$. For fair comparison, ResNet-50 are trained under the same setting. We report classification accuracy on the validation set using 5 repeat runs.

---

[1] All of our experiments were performed using NVIDIA Tesla V100 GPUs with our implementation in PyTorch (Paszke et al., 2017).

### A.2.2 TRANSFERABILITY ON OBJECT DETECTION

We perform experiments on the 80 category COCO detection dataset. We train using the COCO train2017 dataset, and test using the COCO val2017 dataset. Following (Lin et al., 2017), the input image is resized such that its shorter side has 800 pixels. We adopt synchronized SGD training on 16 GPUs. A mini-batch involves 2 images per GPU and 256 anchors per image. We use a weight decay of e-4 and a momentum of 0.9 without dampening. Our fine-tuning is based on $1\times$ setting of the publicly available Detectron (Girshick et al., 2018).

### A.2.3 TRANSFERABILITY ON FACE RECOGNITION

We evaluate and compare the performance of proposed methods as feature extractors for face recognition. The most challenging face identification task on MegaFace dataset (Kemelmacher-Shlizerman et al., 2016) is selected. The MegaFace dataset includes 1M images of 690K different individuals as the gallery set and 100K photos of 530 unique individuals from FaceScrub (Ng & Winkler, 2014) as the probe set. Following (Deng et al., 2019), we select ArcFace as loss, and MS1MV2 (Guo et al., 2016) for training set. For feature extraction, a hidden FC layer is added before the classifier. And the output dimension of FC in the classifier is set equal to the number of individuals of the training set. And the dimension of feature representation is set to 256 for all experiments. We train for 80k iterations with batch size of 512 using 8 GPUs. The initial learning rate is 0.0375. Following (Deng et al., 2019), the learning rate is lowered by 10 times at 30k, 50k and 70k iterations. We use a weight decay of e-4 and a momentum of 0.9 without dampening.

Identification results are given in Table 6, and it should be noted that the Rank-1 accuracy of one million distractors is extremely strict. TopoNets achieve outperforming performances against ResNet-50 with less calculation costs. Comparisons among TopoNets also illustrate the applicability of the optimization for topology in face recognition.

Table 6: Optimization results on MegaFace face identification.

| Network | Params (M) | | Rank-1 (%) |
| --- | --- | --- | --- |
| | Conv/BN | FC | (1M distractors) |
| ResNet-50 | 23.51 | 25.69 | 94.69 |
| *Residual*, $l = 2$ | 20.64 | 12.85 | 96.10(+1.41) |
| *Complete*, $\boldsymbol{\alpha}$ | 20.64 | 12.85 | 96.44(+1.75) |
| *Complete*, $\boldsymbol{\alpha}$, *uniform sparse* | 20.64 | 12.85 | 96.51(+1.82) |
| *Complete*, $\boldsymbol{\alpha}$, *adaptive sparse* | 20.64 | 12.85 | 96.62(+1.93) |

### A.3 REIMPLEMENTATION AND COMPARISION WITH RANDOMLY-WIRED NETWORKS

Since Xie et al. (2019) has not released their codes, we conduct comparisons using TopoNet through replacing graphs to form networks. The three graph generators of ER, BA, WS are performed using NetworkX [2]. with the best configs in their paper. For fair comparison, we do not adopt droppath and dropout for all experiments. The results are given as follows using 5 repeat runs. These results further prove the effectiveness of our optimization method over random baselines.

Table 7: Comparasion with randomly-wired neural networks.

| Topology | Top-1 Acc (%) |
| --- | --- |
| ER (P=0.2) | $77.76 \pm 0.23$ |
| BA (M=5) | $78.08 \pm 0.17$ |
| WS (K=4, P=0.75) | $78.19 \pm 0.25$ |
| our method | $78.60 \pm 0.06$ |

---

[2] https://networkx.github.io/documentation/stable/tutorial.html

## A.4 OPTIMIZED TOPOLOGIES OVER MULTIPLE RUNS

We give results of multiple runs in Fig 5.

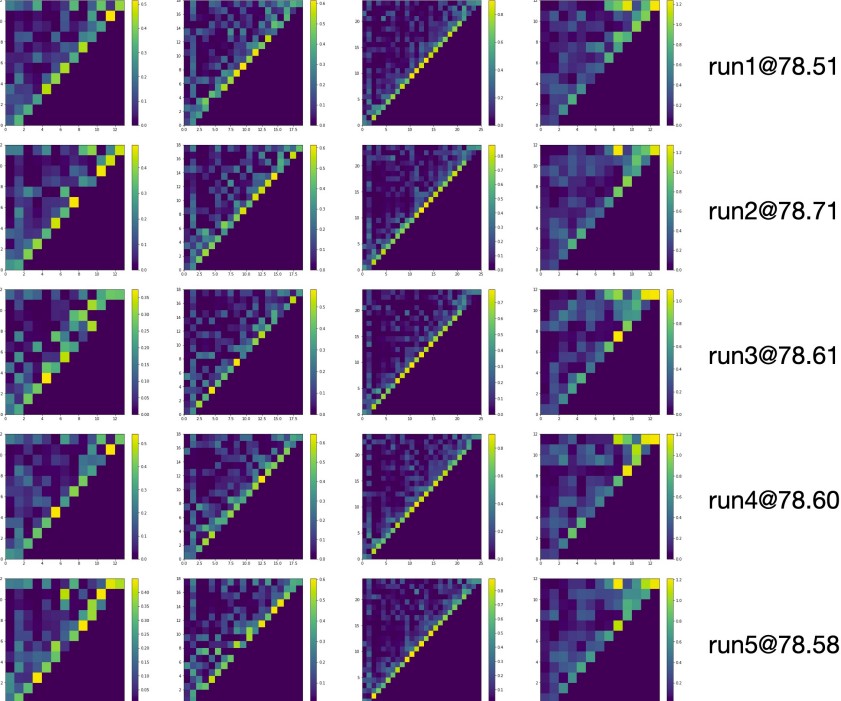

run1@78.51

run2@78.71

run3@78.61

run4@78.60

run5@78.58

Figure 5: We run several times and analyze the optimized topologies, which achieve top-1 accuracy of $78.60 \pm 0.06\%$. The matrices optimized from multiple runs have high similarity, where weights on the diagonal are larger, and connections on the off-diagonal are relatively sparse. This indicates nodes with adjacent topological orderings have more information interactions. Besides, some essential long-range connections are kept as inputs to intermediate layers. Many nodes contribute to the final output, leading to better representations for the subsequent layers.

## A.5 STAND-ALONE ACCURACIES DURING THE SEARCH PROCESS

We give the stand-alone accuracies in Fig. 6.

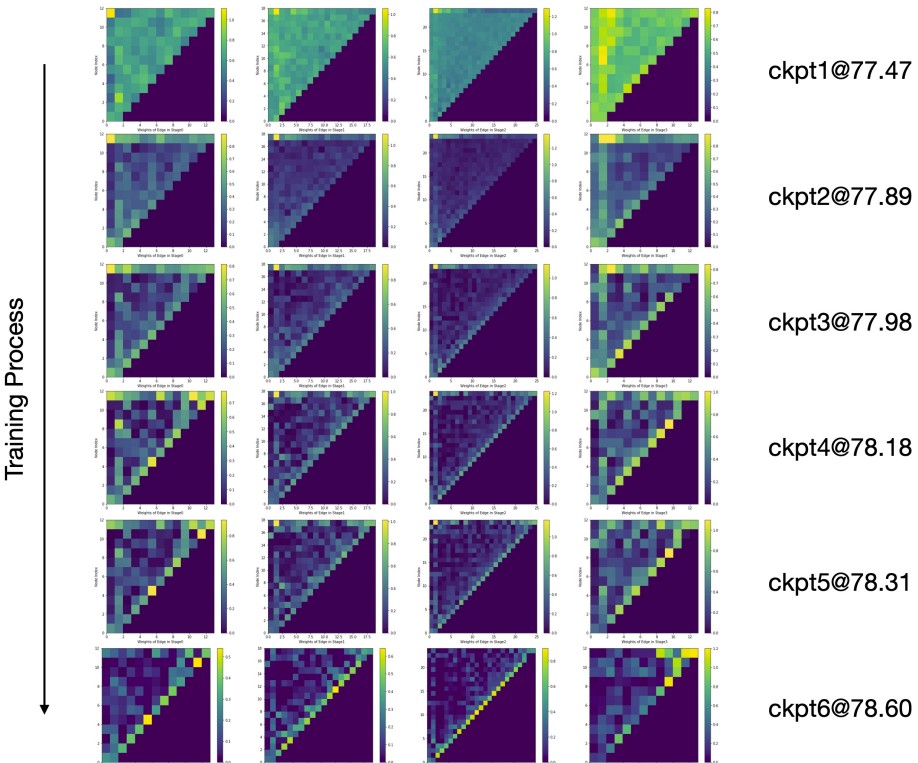

Figure 6: Stand-alone top-1 accuracies on the validation set during optimization. We select topologies under different iterations and freeze their $\boldsymbol{\alpha}$. This reveals the process of topology changes. Then stand-alone accuracies are obtained by training the parameters of network $\mathbf{w}$ from scratch. It is clear that the expression ability of topology itself increases with the training process.

