# OpenReview forum: "Diving into Optimization of Topology in Neural Networks"
_ICLR.cc/2020/Conference — Reject_

### Official Review · AnonReviewer1 · 2019-10-23
**Official Blind Review #1**

**Rating:** 3

**Review:**

The authors proposed a method to optimize the topology of neural networks in a soft fashion. The main idea is to formulate the network as a complete graph (or a sequence of complete subgraphs), and to optimize the relative importance of each edge using gradient descent. The overall approach is similar to differentiable architecture search, except that (1) the continuous architecture is optimized wrt the training set (instead of the validation set), and (2) the learned architecture is never discretized at the end of training.

The paper is well-organized and easy to follow. The authors have also conducted controlled experiments to convincingly show that the method is leading to improvement.

I'm a bit concerned about the technical novelty, however, as the approach can be viewed as an application of (a simplified version of) differentiable NAS to a search space analogous to the one used in [1]. In fact, the notion of soft topology has already been introduced in this prior work (Figure 2 in [1]: "The aggregation is done by weighted sum with learnable positive weights w0, w1, w2"), which was also optimized using gradient descent. A difference between this work and [1] is that whether the underlying graph is complete or randomly generated, but such a distinction is minor (we can always get densely connected random graphs by adjusting the hyperparameters of the graph generator).

In addition, I'm not sure whether the learned continuous \alpha can be conveniently referred to as "topology". Note the mathematical definition of topology is discrete by nature. I believe the authors would need to either revise this terminology (e.g., by referring to it as “soft topology”, as a generalized definition of hard topology), or provide a way to induce discrete subgraphs from the continuous architecture. Sparsity regularization alone may not be sufficient as the non-zero \alpha's are still real-valued.

Minor issue:
I like Figure 1 a lot. However, it seems the equivalence between the 3rd and the 4th sub-figures in Figure 1 can only be established for ResNet-V2 blocks, where there is no ReLU after each addition. It is not immediately obvious how this analysis can generalize to ResNet-V1 blocks (which still offers reasonably good empirical performance).

[1] Xie, Saining, et al. "Exploring randomly wired neural networks for image recognition." arXiv preprint arXiv:1904.01569 (2019).

**Experience Assessment:**

I have published one or two papers in this area.

**Review Assessment: Checking Correctness Of Derivations And Theory:**

I carefully checked the derivations and theory.

**Review Assessment: Checking Correctness Of Experiments:**

I carefully checked the experiments.

**Review Assessment: Thoroughness In Paper Reading:**

I read the paper thoroughly.

---

> ### Author Response · Authors · 2019-11-14
> **Response to Review #1**
>
> We thank the reviewer for the valuable feedback. In the following, we attempt to address the reviewer's concerns:
>
> A1: About the differences with randomly wired neural networks.
> What we are trying to explore is completely different with Randomly Wired Networks.
> Randomly wired networks claimed that random graphs generated by well-defined graph generators are good enough. To its credit, it provides a good platform to explore more flexible network architectures. However, its performance is largely affected by randomness as shown in Fig. 3 in their paper. For the same generator, performances range from 72.6 to 73.4 (ER), 70.7 to 73.2 (BA) and 72.1 to 73.8 (WS). So hyperparameters of generators need to be searched by trial-and-error. Particularly, their experiments with ER generators indicate denseness leads to performance degradation, e.g. 72.6(P=0.8), 72.7(P=0.6), 72.8(P=0.4), 73.4(P=0.2). We also tested ER(P=1.0) (noted Complete with $\alpha$ in Tab. 3) in our paper, resulting in 0.40 lower than our optimization method. These limit the search space in larger ones.
>
> Starting from the residual connection, we point a new topological view to find the reasons for its success. It inspires for dense and important connections. Furthermore, we analyze the influence of different topologies on optimization process, including not just random, but also residual and complete ones. Searching from the complete graph with sparsity constrain proves topology can be optimized rather than randomly wired. Our methods also compatible with existing networks.
>
> We also perform comparision with randomly wired networks as shown in Response to Review #3 A2.
>
> Topology(ImageNet)	Top-1 Acc(%)
>
> ER (P=0.2)			77.76±0.23
> BA (M=5)			78.08+0.17
> WS (K=4, P=0.75)		78.19±0.25
> our method			78.60±0.06
>
> A2: The definition of topology.
> Our definition of topology is consistent with weighted graph in which a number (the weight) is assigned to each edge.
>
> A3: The generalization of the proposed topological view.
> Thank you for your attention to our proposed perspective. The topological view can be used to represent ResNet, ResNeXt, MobileNet-V2 and other networks with residual connections. When there is ReLU after addition such as ResNet-V1, it can be merged into the subsequent node with a form of ReLU/Conv/BN, and the edges should conduct additional inplace ReLU. This transformation is equivalent to original expression, and does not change  the properties of a densely-connect network.

---

### Official Review · AnonReviewer3 · 2019-11-01
**Official Blind Review #3**

**Rating:** 6

**Review:**

*UPDATE* I have read the other reviews, authors' comments and the revised version of the manuscript. I have modified my rating to accept. The updated version, with variance across runs reported, comparison to randomly wired networks, and clearer writing, is substantially better. The core idea is simple to understand in retrospect, and could lead to more follow-up work in the vein of DenseNets and DART (with a more constrained search space).

The paper proposes a refinement of the idea behind DenseNets -- rather than summing over all previous layers' outputs, sum a weighted combination instead where the weights are learned. This idea can be extended to search through the space of all possible residual connections, which they call TopoNet. This is practically achieved by enforcing a sparsity constraint on the learned weights. There is an additional nuance when enforcing the sparsity constraint: downstream layers have many more incoming residual connections, and may need an appropriately scaled sparsity penalty.

 It may help to clarify in the text that weights can be positive or negative (the current motivation from the point of view of residuals at different intervals suggests all the weights should be non-negative).
Table 3 is baffling. Were the initial edge values (column 2) chosen so as to make the number of params and FLOPS somewhat comparable across different rows? How were these initial edge values set (seems very specific for N_3 to go from 46 to 103 when residual interval goes from 4 to 2, etc.)? The comment that number of params and FLOPS changes are negligible is puzzling; clearly Random, p=0.01 should use much fewer FLOPS than Random, p=0.99.
Without additional information behind the numbers for the baselines in Table 3, it is unclear if TopoNets indeed give an improvement over the baselines (Random and Residual).
The text will also benefit from a careful elaboration of the differences in the Random baselines in the paper vs. the Xie et al approach of trying random architectures.

**Experience Assessment:**

I do not know much about this area.

**Review Assessment: Checking Correctness Of Derivations And Theory:**

I assessed the sensibility of the derivations and theory.

**Review Assessment: Checking Correctness Of Experiments:**

I did not assess the experiments.

**Review Assessment: Thoroughness In Paper Reading:**

I made a quick assessment of this paper.

---

> ### Author Response · Authors · 2019-11-14
> **Response to Review #3**
>
> We thank the reviewers for your attention to the details.
>
> A1: Parameters and FLOPs in Table 3.
> We are sorry for not clarifying the changes of params and FLOPs among different topologies. In Table3, for fair comparison, the generated random graph is required to have at least one input and output edge for each node. Otherwise, the graph will be regenerated until the condition is met. Under this setting, different networks share similar params and FLOPs of convolution operations, and the only difference is the number of edges. This ensures that the improvement is not the result of additional computation but topology changes. Without this constrain, nodes without input or output edges can be removed, resulting in fewer params and FLOPs. As shown in Sect. 3.2.3, the removal of nodes for optimized topologies is used to acceleration.
>
> A2: Comparation with random architectures.
> Since [1] has not released their codes, we conduct comparisons using TopoNet through replacing graphs to form networks. The three graph generators (ER, BA, WS) are performed using NetworkX [2] with the best configs in their paper. For fair comparison, we do not adopt droppath and dropout for all experiments. The results are given as follows using 5 repeat runs. These results and Table 3 further prove the effectiveness of our optimization method over random and residual baselines.
>
> Topology(ImageNet)	Top-1 Acc(%)
>
> ER (P=0.2)			77.76±0.23
> BA (M=5)			78.08+0.17
> WS (K=4, P=0.75)		78.19±0.25
> our method			78.60±0.06
>
> [1] Xie, Saining, et al. "Exploring randomly wired neural networks for image recognition." ICCV, 2019
> [2] https://networkx.github.io/documentation/stable/tutorial.html

---

### Official Review · AnonReviewer5 · 2019-11-01
**Official Blind Review #5**

**Rating:** 6

**Review:**

-------------------------      Update after rebuttal    ---------------------------


Thank you for addressing my concerns. I feel the rebuttal did improve the paper, e.g., the significance of results can be evaluated better now. I still like the overall idea of the paper as optimizing connectivity patterns in architectures has so far mostly been ignored while it is actually straight-forward to do (as shown in this work).  I increased my score accordingly. However, the novelty and significance of this work is still limited in my option and therefore I do not argue heavily in favor of accepting this submission.


--------------------------------------------------------------------------------------------------------------------------------------------------




The authors propose a method for learning the connections/ connectivity pattern (dubbed: the topology; meaning which layer is directly connected to which other layer) in neural networks. They do so by weighting connections between layers (e.g., by weighting skip connections) with a real valued parameter. This real-valued parameterization of the connections is then optimized by gradient descent along with the weights of neural networks. The authors also propose L1 regularization on the connectivity parameters to induce sparsity. The proposed method is evaluated by optimizing the topology for ResNets, MobileNetsV2 and their proposed “TopoNets”.

Originality and significance. The manuscript addresses an interesting problem: while there has been lots of work on manually designing better architectures as well as automated design (a.k.a. neural architecture search, NAS), there is little work on optimizing the overall topology (meaning the connectivity patterns between layers). Most prior work solely focuses on search for blocks or cells and then stacking these cells in a pre-defined, not-optimized manner. However, there has been some work also including this in architecture search (e.g., [1,2,3]), and especially the work [4] seems very related but is not discussed. The authors of [4] propose, very similar to this submission, a gradient-based optimization of the connections (in a different way though). The proposed method for optimizing the topology is also very similar to DARTS, simply applied to the connectivity pattern rather than on the operations-level. However, here the topology is optimized along with the network’s weights on the training data rather than the bi-level optimization from DARTS, where the architectural parameters are optimized on the validation data instead (which is very reasonable as one usually considers the architecture as a hyperparameter). I wonder if this has also been considered/tested by the authors of this paper as I would consider the topology to be a hyperparameter which should not be optimized on training but rather validation data. Knowing [4] and DARTS, the proposed method seems to be rather incremental and straightforward rather than ground breaking. While the proposed method allows for more flexible topologies, it introduces different “stages” for their TopoNets, which are actually a similar concept as blocks or cells from the NAS literature. This again does not allow connections between arbitrary layers (but rather only between layers in the same stage; to the best of my understanding). Empirical results show rather small improvements and their significance is unclear (see next paragraph).
Clarity and quality. The paper is mostly well written, well motivated and easy to follow. The mathematical formalism is a little vague at some points (e.g., in Section 2.1., the notation G is used for defining a graph and later in Equation (2) as a function computing feature maps in networks). While the literature on manual design of architectures is thoroughly reviewed, there is missing related work in the context of neural architecture search, as already discussed above. The quality of the results is questionable as differences in accuracies are in almost all experiment rather small (e.g., tuning MobileNetsV2 on Imagenet: 72.62% (original) Top-1 accuracy vs. 72.84% (optimized) and it seems that the authors do only report results for a single run of experiments. In order to assess if the differences are actually statistically significant, the authors would need to report several runs and would need to state, e.g., means and standard deviations.

Overall, the authors address an interesting problem, which seems to have fallen into oblivion in current NAS literature: while researcher optimize cells, which are then stacked to build the final model, not many researcher look into connectivity patterns / topology on the macro level, meaning connections across cells and how cells should be stacked. This paper addresses this problem to some extent. However, the proposed optimization method is, in my opinion, rather incremental (with respect to DARTS and [4]) and therefore of limited novelty and significance. It is currently hard to assess the empirical results due to rather small improvements and the lack of repeated runs of experiments. Mainly for these two reasons, I do not recommend the paper for acceptance.



[1] Esteban Real, Sherry Moore, Andrew Selle, Saurabh Saxena, Yutaka Leon Suematsu, Quoc V. Le, and Alex Kurakin. Large-scale evolution of image classifiers. ICML, 2017.
[2] Thomas Elsken, Jan Hendrik Metzen, and Frank Hutter. Efficient multi-objective neural architecture search via lamarckian evolution. ICLR, 2019.
[3] Hieu Pham, Melody Y. Guan, Barret Zoph, Quoc V. Le, and Jeff Dean. Efficient neural architecture search via parameter sharing. ICML, 2018
[4] Karim Ahmed and Lorenzo Torresani. Maskconnect: Connectivity learning by gradient descent. ECCV, 2018




**Experience Assessment:**

I have published in this field for several years.

**Review Assessment: Checking Correctness Of Derivations And Theory:**

N/A

**Review Assessment: Checking Correctness Of Experiments:**

I assessed the sensibility of the experiments.

**Review Assessment: Thoroughness In Paper Reading:**

I read the paper at least twice and used my best judgement in assessing the paper.

---

> ### Author Response · Authors · 2019-11-15
> **Response to Review #5**
>
> We thank the reviewer for detailed review.
>
> A1: Differences with MaskConnect and DARTS.
> MaskConnect treated original residual networks as being connected only to the immediately preceding module. However, in our proposed topological view, due to the existing of identity mappings, each residual block has already built densely-connected links with all preceding modules. We tried to explain the reason why ResNet work from a new perspective. Additionally, MaskConnect learned to connect each module to K previous modules, where the connections are binary of {0,1}. Fig. 2 in their paper proposed that a very low or very high K yields lower accuracy, which results in the need to select the appropriate hyperparameter for different networks with different depths. In our method, we do not set a fixed in-degree for different layers/nodes. Instead, a complete graph is built and the connections between nodes are continuous. The number of in-degree and the importance of connections are learned adaptively and motivated by task-related loss and sparsity constrains.
>
> Different from DARTS, we define a type-consistent search space to explore the topology-induced impact. Due to memory limitation, DARTS can only search blocks in small datasets, and repeated the learned block to form networks for larger datasets. This suffers from the transfer gap. Our optimization is directly applied to target tasks, e.g. classification, recognition and detection, which guarantees the consistency of search and application.
>
> A2: Whether to adopt alternative optimization strategies.
> There is no theoretical analysis can prove that alternative optimization is better than simultaneous one. DARTS claimed alternative training can ease overfitting in CIFAR-10. But we tested these two types in ImageNet. The top-1 accuracy difference is within $\pm 0.1\%$ in multiple runs. To ensure the consistency of optimization objective, we finally selected the simultaneous one. To further verify the relationship between $\mathbf{\alpha}$ and $\mathbf{w}$. We test the stand-alone accuracies for different topologies during optimization and give the results in Fig. 6.
> The expression ability of topology itself increases with the optimization process on the validation set. Joint training does not lead to structural overfitting on the training set.
>
> A3: About cross-stage optimization.
> Extending optional connections to different stages can expand the search space. But additional transformations are needed to bridge the changes of channels and feature maps for each node with other stages. This contradicts our definition that the down-sample operation is performed by the first node in topological ordering. However, this is an interesting and open question in the future work.
>
> A4: Mathematical formalism correction.
> We use $\mathcal{T}(\cdot)$ to denote the mapping function computing feature maps of $\mathcal{G}$.
>
> A5: Some missing related work is added.
>
> A6: Improvements of our proposed optimization method through multiple runs.
> We provide standard deviations and means in Table 4 through multiple runs. We further apply our method to different networks in more dataset. The improvements are given in Table 2, and Table 3. This shows that the improvements of our method increase with the depth of networks.
>
> Network (CIFAR-100)	origin	optimized	gain
> ResNet-20			69.01	$69.91\pm 0.12$		0.90
> ResNet-32			72.07	$73.34\pm 0.09$		1.37
> ResNet-44			73.73	$75.60\pm 0.14$		1.87
> ResNet-56			75.22	$76.90\pm 0.03$		1.68
> ResNet-110			76.31	$78.54\pm 0.15$		2.23
>
> Network (ImageNet)	depth	origin	optimized	gain
> MobileNet-1.0		53		72.62	$72.86\pm 0.13$		0.24
> MobileNet-1.0-2N	104		75.93	$76.40\pm 0.05$		0.47
> MobileNet-1.0-4N	206		77.33	$77.87\pm 0.09$		0.54
> MobileNet-1.0-6N	308		77.61	$78.36\pm 0.14$		0.75

---

### Official Review · AnonReviewer4 · 2019-11-02
**Official Blind Review #4**

**Rating:** 3

**Review:**

I. Summary of the paper

This paper describes a principled strategy for searching for the most
suitable neural network architecture out of a particular class of
architectures. Specifically, the problem is framed as an optimisation
problem over a set of directed acyclic graphs (DAGs) that correspond to
potential network architectures. By optimising the edge weights of this
representation, a suitable architecture can be generated.
In addition to the aforementioned optimisation scheme, the paper also
presents a regularisation that results in *sparse* networks, i.e.
networks with a smaller number of edges. Multiple experiments on
'tuning' existing architectures on several data sets conclude the paper.

II. Summary of the review

This paper discusses a highly relevant subject, namely how to select
neural network architectures in a principled manner. While the presented
work already goes into a good direction, I cannot give it my endorsement
for acceptance because of the following reasons:

  - The paper is lacking clarity: concepts could be explained somewhat
    better, and the paper is suffering from language/grammar issues that
    make it harder to understand the contents.

  - Lack of experimental or theoretical depth: the proposed method is
    presented as-is; no theoretical analysis of its behaviour is
    performed; while this is not necessarily a problem, as there are
    several empirical experiments, the experimental section is not
    sufficiently detailed: for example, no limitations of the method are
    being discussed and the presented results are not state-of-the-art
    accuracies.

Nevertheless, I want to point out that this paper has the potential to
become an important contribution to the community; it is absolutely
clear that more principled approaches are required to select network
architectures.

In the following, I will comment on the individual aspects in more
detail.

III. Detailed comments (clarity)

- The abstract could be improved in terms of its logical flow. Instead
  of trying to introduce new terminology (macro/micro etc.) here, the
  abstract should rather state directly that this paper frames network
  architecture selection as an optimisation problem over DAGs.

- The use of topology is slightly non-standard here. What is the meaning
  behind the 'macro' and 'micro' operations? This should be explained
  somewhat better.

- Figure 1 should be extended to show an example of how the depicted
  graphs are described through the terminology mentioned in the paper.
  For example, individual edges or nodes could be highlighted and
  referred to in the text to make the 'mapping' clearer.

- I do not understand how operations such as *addition* are represented
  in the DAG. Ideally, this should also be elucidated by a figure.

- When discussing 'intervals' of residual connections, I am assuming
  that the paper refers to how many layers are skipped? If so, this
  should be mentioned and defined explicitly.

- The term 'searching space' should be replaced by 'search space', as
  the latter is more standard usage.

- I do not understand why the optimisation of the topology can decrease
  the computational burden, as claimed on p. 2. The optimisation process
  still has to be performed, just like the training of the network,
  correct? Am I misunderstanding this?

- The caption of Figure 2 could be extended; does a single node type
  mean that the complete network only consists of nodes of that type?
  Moreover, accuracies/errors should be shown in addition to the loss
  curves.

- The term 'dense connection' is vague; I think the paper should use
  'densely-connected graph' here.

- The sentence 'Among these nodes, [...]' refers to the *whole* network,
  and not to the way the output tensor $\mathbf{x}_i$ is processed. Am I
  understanding this correctly?

- I do not understand the initial sentences in Section 2.2; what is the
  meaning of 'cell' in this case?

- The term 'topological structures' should be renamed to '(sub)graphs'
  in order to improve clarity.

- I do not understand the comment on sparsity in Section 2.4. How are
  'moderate sparsity' and Figure 2 connected?

- In the algorithm, I would use 'Sparsity Type' instead of 'Sparse Type'
  to refer to the parameter.

- What does 'Complete' (without $\mathbf{\alpha}$) mean in Table 3?

- The footnote below Table 3 is not referenced anywhere in the text or
  in the table.

- In Figure 3, are the adjacency matrices consistent? What happens if
  the training process is repeated? It would be highly interesting to
  show 'averaged' matrices over multiple runs.

IV. Detailed comments (experiments & theory)

- A theoretical analysis of the proposed method would be interesting.
  Does the optimisation always converge? Are minima unique? What is the
  computational complexity?

  At least some of these aspects should be discussed.

- The limitations of the proposed method are not explained. For example,
  what is the meaning of the sentence on p. 2 about 'excluding the
  influence of the mixture of different layers/nodes'?

  It is my understanding that the proposed method can only change the
  *connections* between blocks of a network, but not the type of layers.
  Is this correct? If so, it would be a major limitation that should be
  mentioned explicitly.

- Another limitation that is not discussed is the scaling to very deep
  networks. How problematic is it to model all potential connections in
  such a network? Are there limits to the current optimisation scheme?
  This needs to be assessed in the experimental section.

- For all experimental tables, standard deviations and means should be
  provided. This is necessary in order to assess the stability of the
  proposed method, because there are multiple sources of stochasticity:
  one arising from the optimisation procedure, the other one arising
  from the training of the network itself.

- The results reported for the experiments are somewhat behind the
  state-of-the-art in terms of accuracy values. This should be stated
  more clearly; I assume that it is caused by limitations of the
  proposed method, which prohibit an application to very recent
  architectures. Is this correct? If so, it should at least be
  mentioned.

- The claim that nodes at the start of a topological ordering contribute
  more to specific stages needs to be (empirically) proven.

V. Style issues

The paper is not easy to read because of several non-standard phrases or
expressions.

- The phrase 'in topology' is often added to a sentence where it does
  not entirely make sense. For example, '[The] architecture can be
  expressed as a directed acyclic graph (DAG) in topology'. I do not
  see the necessity of adding 'in topology' here. There are other places
  at well from which I would remove this phrase.

- 'effective networks' --> 'effective network architectures'

- 'largely affects' --> 'largely affect'

- 'Motivated by which' --> 'Motivated by this'

- 'innovative method' --> 'method' (or 'novel method')

- 'as a complete graph, through' --> 'as a complete graph, and through'

- 'auxiliary sparsity constraint' --> 'an auxiliary sparsity constraint'

- 'named as TopoNet' --> 'called TopoNet'

- 'At initial periods' --> 'Previously' (I am not sure I understand
  this correctly)

- 'red signs' --> 'red arrows'

- 'for its topology' --> 'in terms of its topology' (?)

- 'both combining' --> 'both a combination' (?)

- 'number of interval' --> 'number of intervals'

- 'straight connected' --> 'directly connected'

- 'conduct transformation' --> 'performs a transformation'

- What is the meaning of the phrase 'These may cover the influence
  [...]'? Is this a reference to limitations of existing networks?

- 'opted from' --> 'chosen from'

- 'Following two simple design rules' --> 'We follow two simple design
  rules'

- '1000-dimension' --> '$1000$-dimensional'

- 'We raise two ways' --> 'We describe two ways'

- 'consited' --> 'consisted' / 'consists'

- 'deepen the depth' --> 'increase the depth'

- 'origin' --> 'original'

- 'promotions' --> 'improvements'

- 'can make more profit' --> 'can be useful to improve performance' (I
  am guessing this from the context)

- 'sparseness on representation' --> 'sparsity'

- 'Adaptive one' --> 'The adaptive one'

- 'At the fore' --> 'At the beginning/start'

- I do not understand the sentence about the 'free lunch'. Does it refer
  to the fact that some connections can still be removed from the
  network without decreasing accuracy?

- 'less computation costs' --> 'lower computation costs'

- 'shortcut offers' --> 'shortcuts offer'

- 'benefits optimization' --> 'benefit optimization'

- 'feasible way to the optimization' --> 'feasible way for the optimization'

- Some references in the bibliography are not capitalised consistently

**Experience Assessment:**

I have read many papers in this area.

**Review Assessment: Checking Correctness Of Derivations And Theory:**

I carefully checked the derivations and theory.

**Review Assessment: Checking Correctness Of Experiments:**

I carefully checked the experiments.

**Review Assessment: Thoroughness In Paper Reading:**

I read the paper thoroughly.

---

> ### Author Response · Authors · 2019-11-15
> **Response to Review #4**
>
> We thank the reviewer for the detailed review and suggestions.
> Q1: What is the meaning behind the 'macro' and 'micro' operations?
> A1: In our paper, micro operations denote stacked conv, bn, or activation modules in a layer. Macro one is used to represent connections between layers. This representation is also used in other literature [1,2].
>
> Q2: ... 'intervals' of residual connections…?
> A2: Yes. The number of interval represents the number of layers that make up a residual block.
>
> Q3: … can decrease the computational burden …
> A3: Different from sample-based search method, in our method, the topology and parameters of neural networks are optimized jointly, resulting in small computation cost. The proposed weights in edges have little effect on total params and FLOPs. So we claimed this does not introduce additional computing burdens.
>
> Q4: In Fig. 2, ...single node type mean ... consists of nodes of that type? And the relationship between 'moderate sparsity' and Fig. 2?
> A4: Yes. And top-1 accuracies are shown in labels in the upper right. As given in p. 2, denser networks can achieve higher performance in most cases, e.g. SepConv with interval of 1 (top-1 80.12%) and MBOP with interval of 1 (top-1 81.10%). For Conv operation, the best results of 80.79% is achieved when interval is 2. These indicate moderate sparsity may benefit the optimization and generalization. So we introduce L1 regularization.
>
> Q5: What does 'Complete' (without \alpha) mean in Table 3?
> A5: This means the topology is fixed to complete graph without learnable weights. It is a baseline which reveals the importance of learnable weights of edges and sparsity constrains.
>
> Q6：In Fig. 3, are the adjacency matrices consistent? Does the optimization always converge? Are minima unique? For all experimental tables, standard deviations and means should be provided.
> A6：We do multiple runs and give the results in Fig. 5 and Fig. 6. We provide standard deviations and means in Table 4. For experiments in Table 2 and 3, more structures are compared with our optimization method. In Fig. 5, the matrices optimized from multiple runs have high similarity, where weights on the diagonal are larger, and connections on the off-diagonal are relatively sparse. This indicates nodes with adjacent topological orderings have more information interactions. Besides, some essential long-range connections are built as input to intermediate layers. For the final output, many nodes have their contributions. It is hard to prove the unique of minima in neural networks. Despite the similarities, there are some minor differences between optimized topologies through multiple runs. But the final performances of them are similar. We infer that there is more than one local minima in the search space.
>
> Q7: The claim that nodes at the start of a topological ordering contribute more to specific stages needs to be (empirically) proven.
> A7: Thanks for your attention. Since the topology is represented by a directed acyclic graph, for a node with topological ordering of i, the generated $x_i$ can be only received by node $j$ (where $j>i$). This causes that features generated by front nodes can participate in feature aggregation as a downstream input. It makes the front nodes contribute more.
>
> Q8: It is my understanding that the proposed method can only change the *connections* between blocks of a network, but not the type of layers. Is this correct?
> A8: Yes. Our optimization is conducted under the same type of layers. It provides a convenient and fair platform to compare the effects caused by topology changes. When extending to multiple type of layers, more factors will affect the optimization process, such as the topological orderings and number of params/FLOPs of different types of layers. They are certainly interesting open questions and can serve as future work. To some extent, our experiments of graph damage can also be a preliminary exploration.
>
> Q9: Scaling to very deep networks and the reason why behind the state-of-the-art networks?
> A9: We further conduct experiments on networks with more layers on CIFAR-100 and ImageNet.
> It can be seen that our optimization method is depth-friendly.
>
> Network (CIFAR-100)	origin	optimized	gain
> ResNet-20			69.01	$69.91\pm 0.12$		0.90
> ResNet-32			72.07	$73.34\pm 0.09$		1.37
> ResNet-44			73.73	$75.60\pm 0.14$		1.87
> ResNet-56			75.22	$76.90\pm 0.03$		1.68
> ResNet-110			76.31	$78.54\pm 0.15$		2.23
>
> Network (ImageNet)	depth	origin	optimized	gain
> MobileNet-1.0		53		72.62	$72.86\pm 0.13$		0.24
> MobileNet-1.0-2N	104		75.93	$76.40\pm 0.05$		0.47
> MobileNet-1.0-4N	206		77.33	$77.87\pm 0.09$		0.54
> MobileNet-1.0-6N	308		77.61	$78.36\pm 0.14$		0.75
> A11: Style issues are corrected in the paper. Thank you for your careful proofreading.
> [1] Atwood J, Towsley D. Diffusion-convolutional neural networks. NIPS. 2016.
> [2] Pérez-Rúa J M, Baccouche M, Pateux S. Efficient progressive neural architecture search. arXiv preprint arXiv:1808.00391, 2018.

---

> > ### Comment · AnonReviewer4 · 2019-11-15
> > **re: Your rebuttal**
> >
> > Thank you for your replies! I still have some questions about your experiments, though: can you comment on the limitations of your method or on its theoretical guarantees? This has not been addressed so far. Moreover, what do you think about adding means and standard deviations to the tables?

---

> > > ### Author Response · Authors · 2019-11-15
> > > **Further Response to Review #4**
> > >
> > > In addition to the effectiveness, there are still some limitations in our work. First, the reasons for the improvements caused by sparsity constrains need further study. Second, whether a universal connection can be abstracted from multiple optimized topologies is of great interesting. But we haven't found it yet. Third, whether the search space can be fully utilized needs further verification. We can not provide theoretical guarantees except numerous experimental evidence.
> > >
> > > Sorry for the late update of the mean and variance, repeated runs took a lot of time.

---

### Author Response · Authors · 2019-11-15
**Paper Update**

1. Multiple runs are performed. We give average results in Table 3 and Table 7. And more networks are optimized using our method in CIFAR100 and ImageNet. The results are given in Table 2 and Table 3.
2. We plot the matrices which present the optimized topologies through multiple runs. And we analyze the similarity and difference between them in Fig. 6 in Appendix A. 4.
3. We analyze the optimization process through selecting topologies during training in Fig. 7 in Appendix A. 5. Stand-alone accuracies is provided to verify that the expression ability of topology itself increases with the training process.
4. Comparation experiments with more topologies in randomly wired networks are given in Appendix A. 3.
5. Style and mathematical formalism issues are corrected, and missing related work is added.

---

### Decision · Program_Chairs · 2019-12-19

**Decision:**

Reject

**Comment:**

This paper proposes an approach for architecture search by framing it as a differentiable optimization over directed acyclic graphs. While the reviewers appreciated the significance of architecture search as a problem and acknowledged that the paper proposes a principled approach for this problem, there were concerns about lack of experimental rigor, and limited technical novelty over some existing works.